# Dual engineering of thermodynamics and kinetics in covalent organic frameworks for separation

Zi-Rui Rao[1,2], Xu-Qin Ran[1,3], Zhi-Quan Li[2], Yanlong Chen[4], Xiu-Ping Yan [1,2] & Hai-Long Qian [1,2] ✉

Dual engineering thermodynamics and kinetics is crucial for achieving high-performance separation, but remains challenging. Here, we pioneer in the engineering of covalent organic framework (COF) from both thermodynamic and kinetic perspectives by rational design of a hollow trifluoromethyl functionalized COF (HTpBPa-F) for enhanced chromatographic separation of halogenated isomers. The trifluoromethyl introduction not only promotes the thermodynamic selectivity for halogenated isomers but also enhances separation kinetics by facilitating formation of a hollow structure. As a result, HTpBPa-F yields higher resolution and column efficiency for pairs of halogenated isomers than either solid fluorinated COF or trifluoromethyl-free COF. Density functional theory calculations reveal thermodynamic selectivity of HTpBPa-F for halogenated isomers results from C-H···π, π-π and dipole-dipole interactions. Molecular dynamics simulations demonstrate high diffusion coefficient of hollow structure leads to low transport resistance, enhancing the kinetics of separation. This work offers insights into simultaneously tailoring COFs from thermodynamics and kinetics for the high-performance separation.

Covalent organic frameworks (COFs) are crystalline porous polymers formed by precise atomic integration of organic units into extended 2D or 3D structures, featuring periodic order structure, precisely designable functions and topology, as well as good stability[1–3]. COFs have already shown great promise as stationary phases in chromatography[4–6]. By modifying COFs with specific functional groups, researchers can tune their thermodynamic properties to achieve separations of homologs, isomers, and chiral enantiomers. For example, fluorine, amino group, glutathione, and chiral biomolecules (amino acids, peptides, and enzymes) have been grafted onto COFs to leverage specific interactions for enhanced chromatographic separation[7–13]. However, such thermodynamic engineering alone is insufficient, as it represents only one of the factors governing chromatographic separation.

It is well known that chromatographic separation is controlled by both thermodynamics and kinetics[14,15]. On the one hand, thermodynamics dictates the equilibrium distribution of analytes between the mobile and stationary phases, defining the inherent resolution of mixed compounds. Parameters such as the capacity factor ($k$) and partition coefficient ($K$) are fundamentally thermodynamic in nature. On the other hand, kinetics controls the equilibrium rate, directly influencing the quality of chromatographic separation such as peak broadening and symmetry. The theoretical plate height ($H$), determined by eddy diffusion, molecular diffusion, and mass transfer resistance, is a key kinetic parameter in chromatography. Therefore, an ideal stationary phase must provide strong thermodynamic selectivity to analytes, while its structure must also facilitate fast kinetics to minimize peak broadening and ensure high-resolution efficiency.

[1]State Key Laboratory of Food Science and Resources, Jiangnan University, Wuxi, China. [2]Institute of Analytical Food Safety, School of Food Science and Technology, Jiangnan University, Wuxi, China. [3]Department of Light Chemical Engineering, Jiangnan University, Wuxi, PR China. [4]School of Pharmaceutical Sciences, Guangzhou University of Chinese Medicine, Guangzhou, Guangdong, China. ✉e-mail: hlqian@jiangnan.edu.cn

Halogenation is a critical process with profound implications for the environment, organisms, and industry[16,17]. For instance, natural organic compounds can be transformed into halogenated disinfection byproducts during chemical water disinfection[18,19]. In organisms, halogenation is an important biosynthetic pathway. A key example is the biosynthesis of iodine-containing thyroid hormones, which are vital for regulating metabolism[20]. Moreover, many industrial productions like vinyl chloride monomers (for polyvinyl chloride plastics) rely on halogenation[21]. During these halogenations, halogen substituents can attach to different carbon atoms, yielding diverse halogenated isomers. Despite sharing the same molecular formula and often very similar physical properties, halogenated isomers can differ evidently in biological activities and chemical behaviors[22–24]. Therefore, the separation of halogenated isomers is not merely a technical challenge but a fundamental requirement in chemical synthesis, metabolic research, and environmental analysis. However, there are currently no commercial columns specifically designed for separating halogenated isomers. Instead, separations rely on broad-spectrum columns such as HP-5.

Herein, we pioneer in the engineering of COF from both thermodynamic and kinetic perspectives as specific stationary phase for high-resolution gas chromatographic separation of halogenated isomers. To this end, a trifluoromethyl containing monomer 2,5-diaminobenzotrifluorid (Pa-F) is synthesized for the purpose of being condensed with 1,3,5-tris(p-formylphenyl)benzene (TpB) to afford a hollow fluorinated COF named HTpBPa-F via Ostwald ripening method. Trifluoromethyl groups are incorporated to modulate thermodynamic selectivity, while the hollow architecture is aimed at controlling the kinetics for halogenated isomers separation. In addition, a solid fluorinated COF (STpBPa-F) and a fluorine-free COF (STpBPa) are prepared as controls to investigate the distinct kinetic and thermodynamic effects of HTpBPa-F on the separation of halogenated isomers. This work demonstrates the accessibility and guidance of simultaneously engineering thermodynamics and kinetics in COFs for high-performance separation.

## Results

### Engineering and preparation of hollow fluorinated COF

Aiming at engineering of thermodynamics and kinetics in COFs for high-resolution separation of halogenated isomers, our rational design is dominantly centered on developing a hollow fluorinated COF. On the one hand, the strong electron-withdrawing effect of the fluorine groups can easily induce high dipole moment, enabling fluorinated COF to promote thermodynamic selectivity for halogenated isomers. On the other hand, the calculated stacking energy of three layers with trifluoromethyl groups ($-64$ kcal mol$^{-1}$) is much higher than that of F-free layer ($-50.9$ kcal mol$^{-1}$) (Fig. 1), indicating that the introduction of trifluoromethyl groups can enhance the interlayer interactions. Such an enhancement facilitates the formation of a hollow-structured COF guided by the Ostwald ripening, which in turn improves the kinetics for separating halogenated isomers. In addition, a fluorine-free COF and a solid fluorinated COF were also prepared as controls to investigate the distinct roles of kinetics and thermodynamics in the separation of halogenated isomers, respectively.

Accordingly, a trifluoromethyl groups containing precursor Pa-F was first synthesized by reacting Pa with CF$_3$SO$_2$Na (Supplementary Figs. S1–S3), then TpB was further condensed with Pa-F to obtain the proposed hollow fluorinated HTpBPa-F via Ostwald ripening in o-DCB/n-BuOH/3 mol L$^{-1}$ HAc (5:5:1, v/v/v, 3.3 mL) at 90 °C for 72 h (Fig. 1). In addition, the fluorine-free COF (STpBPa) was synthesized via the condensation of TpB and Pa in a mixed solution of mesitylene/ethanol/3 mol L$^{-1}$ HAc (5:5:1, v/v/v, 2.2 mL) at room temperature for 72 h. Ostwald ripening is a time-dependent process in which small particles dissolve and redeposit onto large particles, ultimately resulting in the formation of hollow structures[25–28]. The Ostwald ripening formation of hollow tubular HTpBPa-F was explored by monitoring its morphology at different reaction times (Supplementary Fig. S4). A solid tubular structure formed after 12 h of reaction time. Incidentally, this solid-state COF (named as STpBPa-F) was served as control in subsequent separation experiment. As the reaction time increased, the small microcrystals in solid tube started to dissolve and recrystallize on the external surface at 24 h. Further increasing the reaction time to 48 h

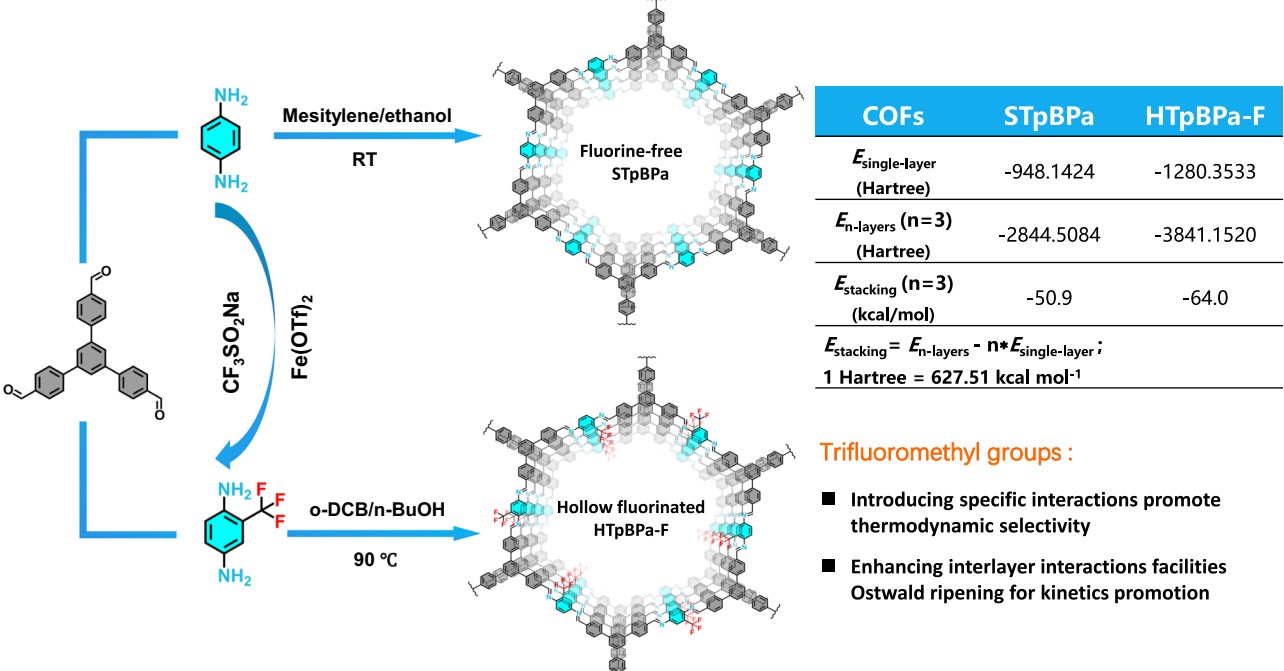

| COFs | STpBPa | HTpBPa-F |
|---|---|---|
| $E_{\text{single-layer}}$ (Hartree) | -948.1424 | -1280.3533 |
| $E_{\text{n-layers}}$ (n=3) (Hartree) | -2844.5084 | -3841.1520 |
| $E_{\text{stacking}}$ (n=3) (kcal/mol) | -50.9 | -64.0 |
| $E_{\text{stacking}} = E_{\text{n-layers}} - \text{n}*E_{\text{single-layer}}$ ; 1 Hartree = 627.51 kcal mol$^{-1}$ | | |

**Trifluoromethyl groups :**

- Introducing specific interactions promote thermodynamic selectivity
- Enhancing interlayer interactions facilities Ostwald ripening for kinetics promotion

**Fig. 1 | Illustration for the synthesis of fluorine-free STpBPa and hollow HTpBPa-F.** The STpBPa and hollow HTpBPa-F were prepared using TpB with Pa and Pa-F, respectively. Incorporation of trifluoromethyl groups not only imp the thermodynamic selectivity of COF for halogenated isomers but also but also enhances interlayer interactions for improved kinetics as evidenced by $E_{\text{stacking}}$.

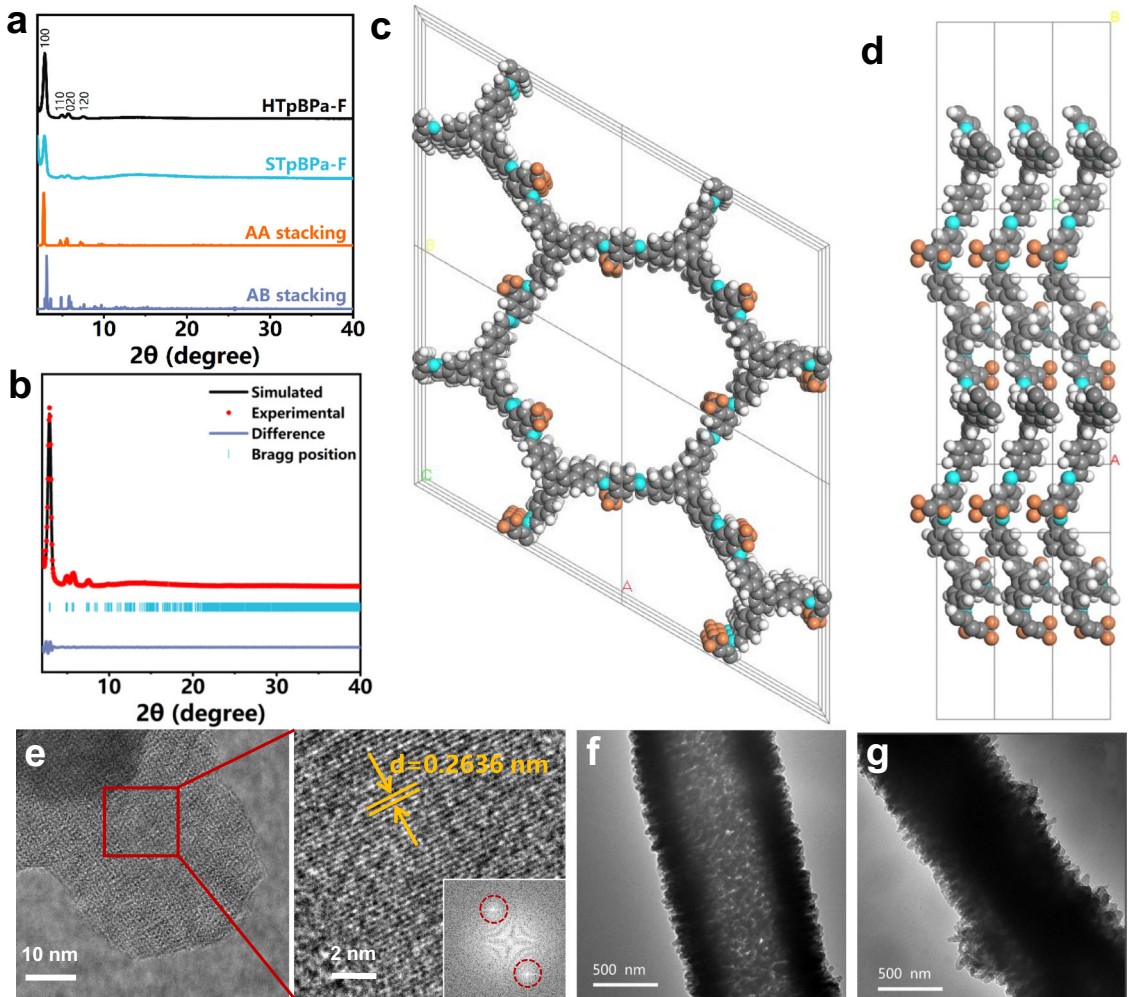

**Fig. 2 | Characterization of fluorinated COFs. a** Experimental and simulated PXRD patterns of HTpBPa-F and STpBPa-F. **b** Pawley refinement results of simulated HTpBPa-F (Rwp 7.37% and Rp 5.48%). **c** Front and **d** side view of AA eclipsed unit cell of HTpBPa-F. **e** Cryo-TEM images of HTpBPa-F. TEM image of **f** HTpBPa-F and **g** STpBPa-F.

produced larger internal cavities in the tubes, but some small particles were still observed inside. The complete hollow tubular structure was finally formed after 72 h. Notably, an excessively long reaction time (120 h) induced the partial collapse of the hollow structure.

### Characterization of HTpBPa-F, STpBPa-F and STpBPa

Crystalline structure of prepared COFs was confirmed with powder X-ray diffraction (PXRD) analysis and computational modeling. HTpBPa-F and STpBPa-F were prepared using the same precursor and reaction conditions but with different reaction times, resulting in no differences in their composition and crystallinity. Both HTpBPa-F and STpBPa-F exhibited a series of PXRD peaks at 2.9°, 5.0°, 5.7°, and 7.6° (Fig. 2a), corresponding to reflection from (100), (110), (020), and (120) planes, respectively. The experimental PXRD patterns matched those simulated from AA stacking model rather than AB stacking model, indicating that both HTpBPa-F and STpBPa-F adopt a two-dimensional AA stacking structure (Fig. 2b and Supplementary Fig. S5). Further refinement produced a specific hexagonal unit cell with space group of P1, $\alpha = \beta = 90°$, $\gamma = 120°$, a = b = 35.9927 Å, and c = 3.7464 Å (Fig. 2c, d and Supplementary Table S1). Cryo-transmission electron microscopy (TEM) revealed the interplanar spacing of 0.2636 µm for HTpBPa-F (Fig. 2e).

The almost identical Fourier-transform infrared (FTIR) spectra of HTpBPa-F and STpBPa-F confirmed their same chemical composition. The presence of characteristic FTIR peaks at 1618 cm$^{-1}$ (C = N

stretching vibration) proved the successful Schiff base reaction (Supplementary Fig. S6). More importantly, the introduction of fluorine groups was confirmed by the appearance of a C−F stretching vibration at 1139 cm$^{-1}$ in the FTIR spectrum, a carbon signal of C-F at 118 ppm in the $^{13}$C solid-state nuclear magnetic resonance (SNMR) spectrum (Supplementary Fig. S7), and an F 1 s peak at 688 eV in the full-scan and high-resolution X-ray photoelectron spectra (Supplementary Fig. S8). Scanning electron microscopy (SEM) images exhibited the tubular morphology of HTpBPa-F and STpBPa-F (Supplementary Fig. S9). Crucially, TEM images revealed a hollow tubular structure of HTpBPa-F in contrast to the solid tubular structure of STpBPa-F (Fig. 2f and g). HTpBPa-F possessed a significantly higher BET surface area (1049 m$^2$ g$^{-1}$) than STpBPa-F (195 m$^2$ g$^{-1}$), but a similar pore size of ~2.18 nm (Supplementary Figs. S10 and S11). Thermogravimetric analysis revealed that HTpBPa-F and STpBPa-F exhibited good thermal stability with minimal weight loss up to 400 °C, indicating their potential as stationary phases for GC (Supplementary Fig. S12).

Similarly, the crystalline structure, composition, and pore properties of fluorine-free STpBPa were also characterized. STpBPa adopt a two-dimensional AA stacking structure and the refined unit cell was featured with space group of P6/m, $\alpha = \beta = 90°$, $\gamma = 120°$, a = b = 35.5650 Å, and c = 3.6275 Å (Supplementary Figs. S13 and S14, Supplementary Table S2). The appearance of C = N at 1618 cm$^{-1}$ in FTIR spectra of STpBPa indicates the successful Schiff base reaction (Supplementary Fig. S15). SEM and TEM images exhibited the solid

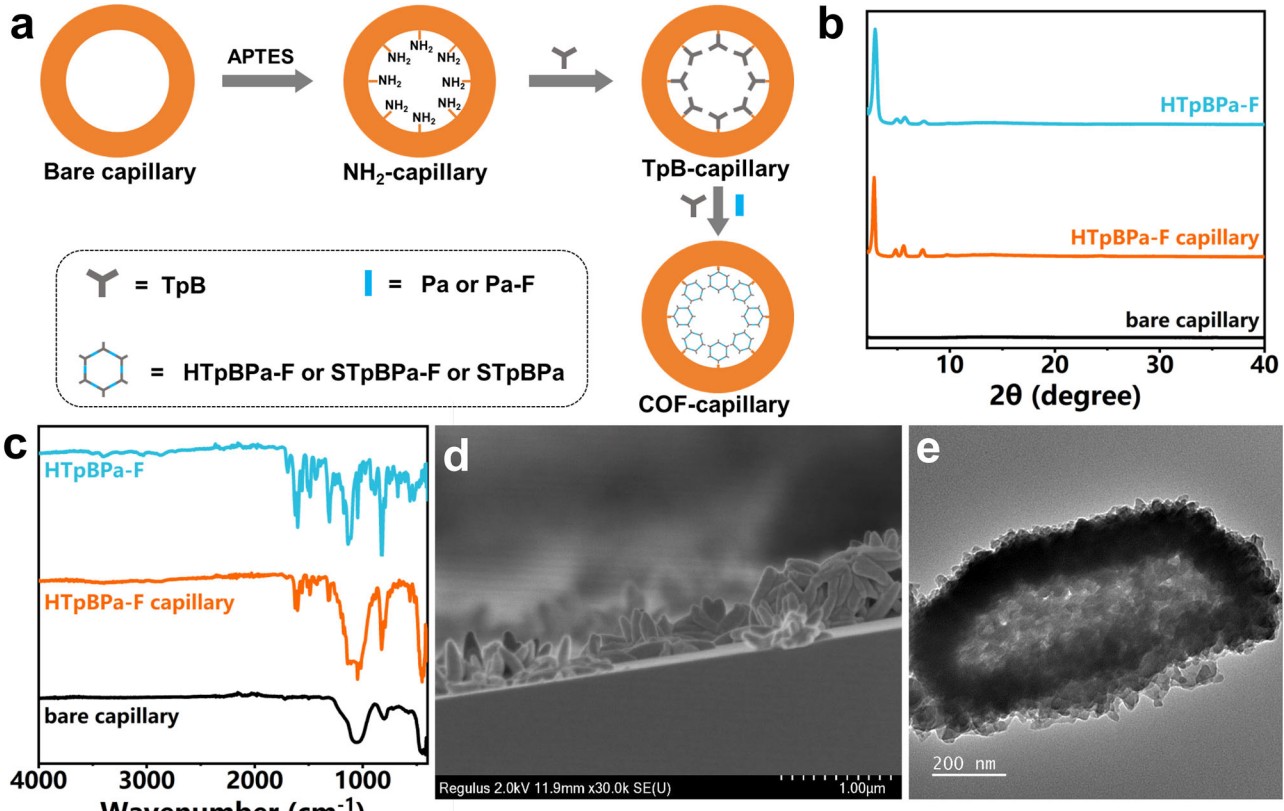

**Fig. 3 | Characterization of COFs bonded capillary columns. a** In-situ growth for preparation of COFs bonded capillary columns. **b** PXRD patterns of bare capillary, HTpBPa-F capillary and HTpBPa-F. **c** FTIR spectra of bare capillary, HTpBPa-F capillary and HTpBPa-F. **d** Cross-sectional SEM images of HTpBPa-F-bonded capillary column. **e** TEM images of HTpBPa-F on capillary column.

spherical morphology (Supplementary Fig. S16). The BET surface area and pore size distribution of STpBPa were calculated to be 130 m² g⁻¹ and ~3.4 nm, respectively (Supplementary Fig. S17). The STpBPa was thermally stable up to 350 °C (Supplementary Fig. S12).

**Characterization of HTpBPa-F, STpBPa-F and STpBPa capillary columns**

We further fabricated capillary columns bonded with the proposed COFs by using an in-situ growth approach for the subsequent separation of halogenated isomers (Fig. 3a). No obvious changes of PXRD patterns indicate the retention of crystalline structure of HTpBPa-F, STpBPa-F and STpBPa after growth on the inner wall of capillaries (Fig. 3b and Supplementary Fig. S18). The prepared three COFs bonded capillaries gave the appearance of evident characteristic peaks for the three COFs in addition to the peak of silica capillaries in their FTIR spectra (Fig. 3c and Supplementary Fig. S19). SEM images clearly showed a smooth inner surface of the bare capillary, whereas the inner walls of the COF-coated capillary columns were uniformly covered with dense particles of HTpBPa-F, STpBPa-F and STpBPa (Fig. 3d, Supplementary Figs. S20, S21a and S22a). These results prove the successful fabrication of COFs-bonded capillary columns. Although the confined space of the capillaries yielded smaller particles of HTpBPa-F, STpBPa-F and STpBPa on the inner wall than those prepared in a conventional tube, TEM images confirmed that HTpBPa-F maintained its hollow structure, while both STpBPa-F and STpBPa were solid architecture (Fig. 3e, Supplementary Figs. S21b and S22b).

**Separation of halogenated isomers**

Baseline separation of halogenated isomers is of great significance for chemical synthesis, pharmaceutical development, and environmental analysis. For example, p-chloroaniline is a regulated and carcinogenic aromatic amine from dye decomposition, but this restriction does not apply to its ortho and meta isomers[29]. In order to explore the separation capability of HTpBPa-F for halogenated isomers, the three synthesized COF-bonded capillaries were applied to separate a series of aliphatic and aromatic halogenated isomers containing fluorine, bromine, or chlorine, namely chloroaniline (CA), fluoroaniline (FA), monochlorobiphenyl (MoCB), 1,2-dichloroethylene (1,2-DCE), dibromopropane (DBP), dichloropropane (DCP), tetrachlorobenzene (TeCB), bromochlorobenzene (BCB), and chloronitrobenzene (CNB).

The HTpBPa-F bonded capillary column gave baseline separation for all the studied isomers with high-resolution (2.3–28.0) and high column efficiency (1137–7389 plates m⁻¹) under constant temperature (Fig. 4 and Supplementary Table S3–S4). In contrast, the STpBPa-F bonded capillary column yielded different retention times for the isomers, but notable overlap of some peaks prevented their baseline separation (Supplementary Fig. S23). The STpBPa bonded capillary column exhibited poor separation with many isomers having identical retention times, resulting in completely overlapping peaks. (Supplementary Fig. S24). The co-elution of CA, MoCB and TeCB isomers, as well as poor resolution of FA, 1,2-DCE and BCB isomers on benchmark commercial column HP-5 further indicate the great potential of HTpBPa-F bonded capillary column as specific chromatography column for halogenated isomers (Supplementary Fig. S25). In addition, the HTpBPa-F bonded capillary column exhibited baseline separation of the homologs including n-alkanes and n-alcohols, highlighting its broad practical utility (Supplementary Fig. S26).

**Separation mechanism**

The molecular sizes of separated isomers range from 0.3 to 1.3 nm (Supplementary Table S5), all of which are smaller than the pore size of HTpBPa-F. As a result, these molecules can diffuse into the pores, and

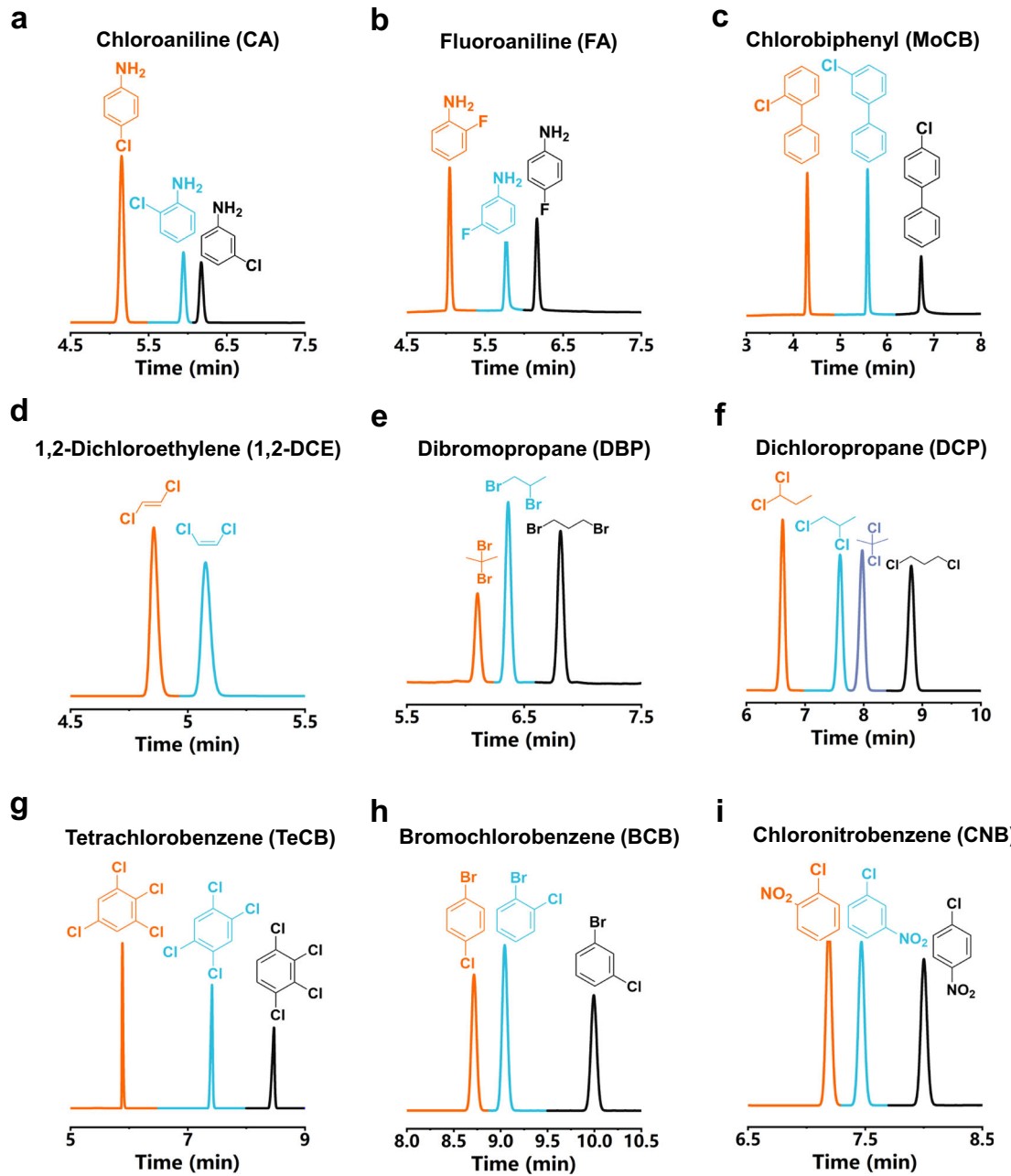

**Fig. 4 | Chromatograms of halogenated isomers on the HTpBPa-F-bonded capillary column. a** CA (150 °C, 1.2 mL min⁻¹ of N₂). **b** FA (150 °C, 1.1 mL min⁻¹ of N₂). **c** MoCB (150 °C, 1.3 mL min⁻¹ of N₂). **d** 1,2-DCE (90 °C, 1.5 mL min⁻¹ of N₂). **e** DBP (90 °C, 1.1 mL min⁻¹ of N₂). **f** DCP (40 °C, 1.2 mL min⁻¹ of N₂). **g** TeCB (150 °C, 1.2 mL min⁻¹ of N₂). **h** BCB (80 °C, 1.5 mL min⁻¹ of N₂). **i** CNB (120 °C, 1.2 mL min⁻¹ of N₂).

the spatial effect of the pores can account for the separation process. To further understand the separation of target halogenated isomers on HTpBPa-F stationary phase, we delved into the separation mechanism from both thermodynamic and kinetic perspectives. According to the separation results, the STpBPa bonded capillary column showed poor resolution for all the studied isomers, with some of them even co-eluting completely (Supplementary Fig. S24). In contrast, the STpBPa-F with same structure but containing tri-fluoromethyl group led to the evident improvement of the separation. All the isomers yielded isolated peaks on the STpBPa-F bonded capillary column, but some were not baseline resolved due to peak broadening and tailing (Supplementary Fig. S23). The partition coefficient ($K$) is an important parameter in chromatographic thermodynamics. So, we compared $K$ of all the isomers on STpBPa-F and

STpBPa bonded capillary column. All isomers exhibit a greater partition coefficient difference ($\Delta K$) on the STpBPa-F bonded capillary column than on the STpBPa column, indicating that the introduced trifluoromethyl groups efficiently promote thermodynamic selectivity (Supplementary Tables S6–S8).

The thermodynamic parameters for separation of halogenated isomers were further investigated (Supplementary Table S9). Negative $\Delta S$ values (−5.0 to −67.2 J mol⁻¹ K⁻¹) imply that all the separated halogenated isomers are 'locked' in a more ordered state while interacting with the HTpBPa-F stationary phase during separation. As is known, enthalpy change ($\Delta H$) stems from the collective effect of multiple noncovalent interactions. The negative $\Delta H$ typically indicates the presence of strong and specific interactions during the separation process[30]. So the negative $\Delta H$ values observed in the chromatographic separation (−12.1 to

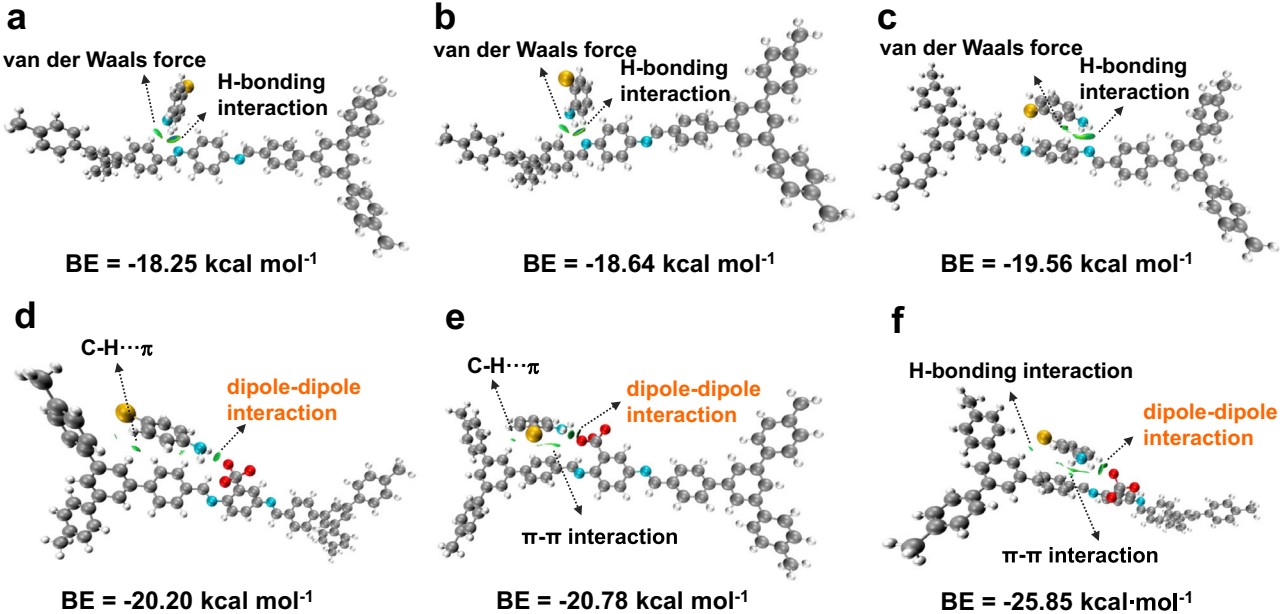

**Fig. 5 | DFT-based IGMH analysis of the interaction between the CA isomer and prepared COFs. a** p-CA and STpBPa, **b** o- CA and STpBPa, **c** m-CA and STpBPa, **d** p-CA and STpBPa-F, **e** o-CA and STpBPa-F, **f** m-CA and STpBPa-F.

−42.9 KJ mol⁻¹) demonstrate that some strong interactions indeed occur between HTpBPa-F and the halogenated isomers.

In order to figure out the specific interactions between prepared COFs and the halogenated isomers, Density functional theory (DFT) calculation was performed using the unit cell of STpBPa-F and STpBPa to represent COFs structure, and CA as model isomer (Supplementary Fig. S27). The calculated binding energies (BE) of *p*-, *o*-, and *m*-CA on STpBPa-F were calculated to be −20.20, −20.78 and −25.85 kcal mol⁻¹, respectively. These values are higher than those on STpBPa (−18.25, −18.64, and −19.56 kcal mol⁻¹), indicating that the trifluoromethyl group can enhance the host-guest interactions. The Independent Gradient Model of the Hartree-Fock analysis (IGMH) was further applied to visualize and explore the specific interactions. As shown in Fig. 5a–c, hydrogen bonding interaction and van der Waals forces were dominantly involved in the separation of CA isomers with STpBPa. In contrast, the interaction between STpBPa-F and CA molecules involved not only C-H⋯π and π-π interactions but also a significant interaction contributed from the trifluoromethyl groups. This additional interaction is attributed to dipole-dipole forces, resulting from the strong electron-withdrawing effect of the fluorine atoms (Fig. 5d −f). Consequently, the introduction of the trifluoromethyl group was further confirmed to thermodynamically favor the separation of the halogenated isomers.

In terms of dynamics, the hollow-structure HTpBPa-F achieved a superior separation of studied isomers compared with STpBPa-F, despite sharing the same composition. The HTpBPa-F bonded capillary column is capable of baseline-resolving all the studied isomers with symmetric chromatographic peaks and high column efficiency, indicating the successful kinetics engineering of hollow structure for separation of halogenated isomers. Van Deemter equation quantitatively describes various kinetic factors in chromatographic columns (See SI for details)[31]. The longitudinal diffusion coefficient (*B*) and mass transfer coefficient (*C*) of STpBPa-F, and HTpBPa-F bonded capillary column in Van Deemter equation was calculated by using the plots of H/u against 1/u² (Supplementary Fig. S28, Supplementary Table S11). The fitted *C* value of HTpBPa-F bonded capillary column is much lower than that of the STpBPa-F, indicating the lower mass transfer on HTpBPa-F than STpBPa-F.

Further molecular dynamics (MD) simulations were conducted to understand the transport behavior of halogenated isomers in HTpBPa-F and STpBPa-F. We also chose CA as the model isomers. Three CA isomers exhibited greater mobility in HTpBPa-F than in STpBPa-F within the 50 ps timeframe (Fig. 6a, b, Supplementary Figs. S29 and S30). According to the mean squared displacement (MSD) versus time curves, the diffusion coefficients (*D*) of *o*-, *m*-, and *p*-CA in HTpBPa-F were 5.88 × 10⁻⁵, 4.58 × 10⁻⁵ and 5.49 × 10⁻⁵ cm² s⁻¹, which were larger than those in STpBPa-F (3.68 × 10⁻⁵, 1.61 × 10⁻⁵ and 1.84 × 10⁻⁵ cm² s⁻¹) (Fig. 6c–e). In chromatographic separation, the mass transfer resistance is inversely proportional to diffusion coefficient and directly proportional to thickness of stationary phase (See SI for details)[32]. Therefore, the large *D* and short transfer pathway of HTpBPa-F led to lower mass transfer resistance on HTpBPa-F, confirming that the hollow structures can regulate the kinetic properties and improve the quality of chromatographic separation.

## Separation stability and repeatability
Separation performance of CA isomers on the same HTpBPa-F column at different time was compared to evaluate the separation stability (Supplementary Fig. S31 and Supplementary Table S12). In contrast to the initial prepared HTpBPa-F bonded capillary column, the column used for 120 days gave reductions of only 0.4−2.6% and 0.1−0.2% in column efficiency (*N*) and capacity factor (*k*), respectively. Moreover, even after the column was subjected to 10 continuous programmed temperature cycles from 50 to 300 °C at 2 °C min⁻¹, the *N* and *k* for the CA isomers showed no significant change, demonstrating the excellent long-term stability of the prepared HTpBPa-F column.

Both the operational consistency and inter-batch precision were further investigated to assess the repeatability of HTpBPa-F bonded capillary column (Supplementary Table S13). Specifically, the relative standard deviations (RSDs) for the retention time and peak area of CA isomers with a same HTpBPa-F capillary column were 0.11−0.12% and 2.8−3.2% for eight consecutive runs, respectively. For five runs repeated across different days, the corresponding RSDs were 0.10−0.11% and 5.8−6.1%, respectively. These results clearly demonstrate the high repeatability of the HTpBPa-F capillary column. In addition, excellent reproducibility was also confirmed by the RSDs of 0.11−0.22%

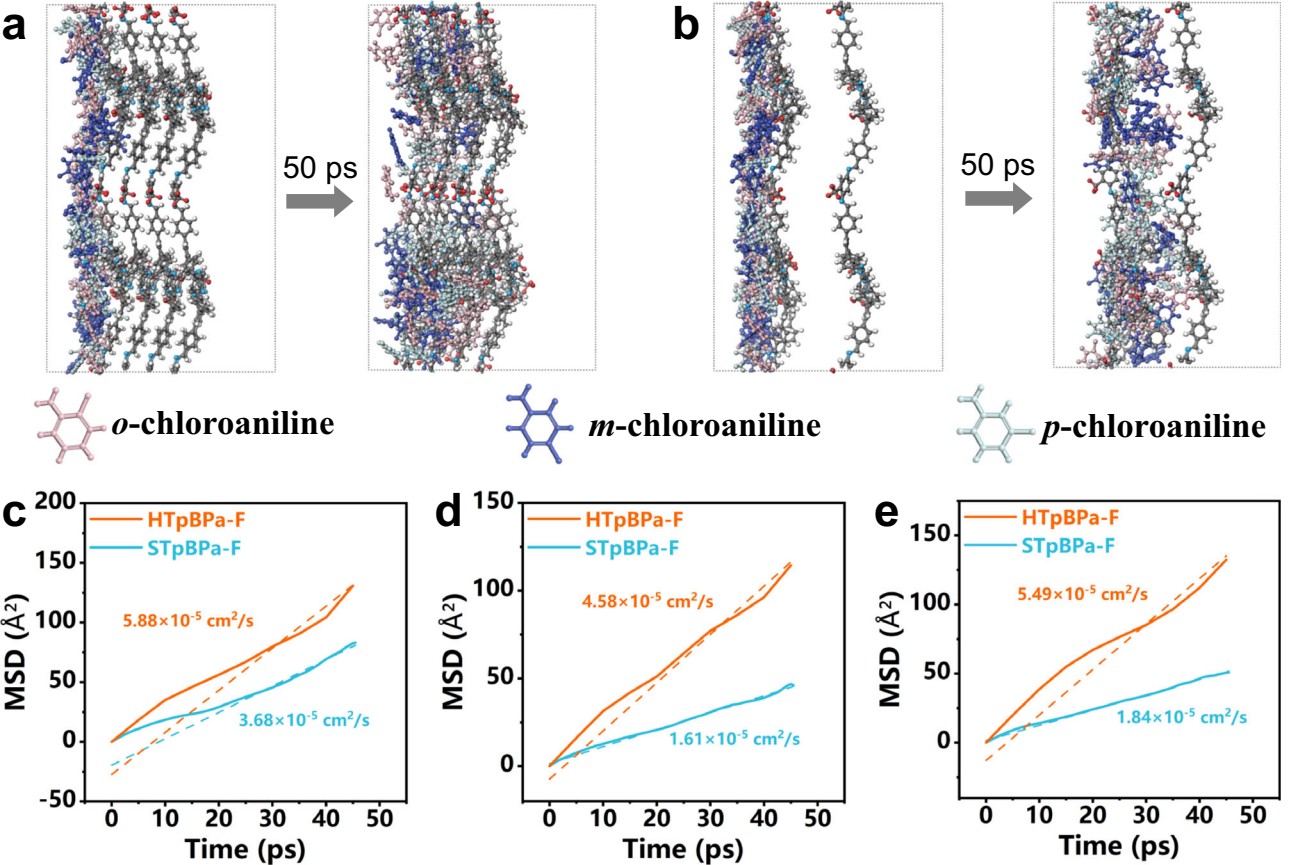

**Fig. 6 | MD simulation of CA isomer in prepared COFs.** Representative Snapshot images of the MSD for CA isomers in **a** STpBPa-F and **b** HTpBPa-F. Time-dependent MSD curves (solid line) and their fitting curves (Dashed line) of **c** *o*-CA, **d** *m*-CA, and **e** *p*-CA in STpBPa-F and HTpBPa-F.

(retention time) and 6.8–7.4% (peak area) for CA isomers obtained using columns from three different batches.

## Discussion

In summary, we rationally designed a hollow fluorinated COF to engineer both the thermodynamics and kinetics for achieving high-resolution chromatographic separation of halogenated isomers. The fluorinated COF provided additional dipole-dipole force in addition to C-H⋯π and π-π interactions to modulation of thermodynamic selectivity, achieving high-resolution of halogenated isomers. The hollow structure enabled the large diffusion coefficient of halogenated isomers, leading to the low transform resistance in kinetics and achieving high column efficiency. This work provides strong support for tailoring COFs from thermodynamics and kinetics for the high-performance separation.

## Methods

### Synthesis of HTpBPa-F and STpBPa-F

The monomer of Pa-F was first synthesized (see SI for details). Then TpB (29.3 mg, 0.075 mmol) and Pa-F (19.8 mg, 0.113 mmol) were mixed in o-DCB (0.4 mL) and n-BuOH (2.6 mL) under sonication for 10 min, then 3 M acetic acid (0.3 mL) was added under sonication for additional 5 min to obtain pre-polymerization solution. After three freeze-pump-thaw cycles, the pre-polymerization solution was reacted at 90 °C for 72 h. The finally obtained precipitate was collected by centrifugation, and washed with THF and EtOH, dried under vacuum at 60 °C for 12 h to afford hollow fluorinated COF HTpBPa-F. The solid fluorinated COF (STpBPa-F) was synthesized under identical process to HTpBPa-F, except that the reaction time was 12 h.

### Synthesis of STpBPa

Typically, TpB (17 mg, 0.04 mmol) and Pa (6.5 mg, 0.06 mmol) were mixed in mesitylene (1 mL) and ethanol (1 mL) under sonication for 10 min, then 3 M acetic acid (0.2 mL) was added under sonication for additional 5 min to obtain pre-polymerization solution. After three freeze-pump-thaw cycles, the pre-polymerization solution was reacted at room temperature for 72 h. The finally obtained precipitate was collected by centrifugation, washed with THF and EtOH, and dried under vacuum at 60 °C for 12 h to afford fluorine groups-free COF (STpBPa).

### Preparation of HTpBPa-F, STpBPa-F and STpBPa bonded capillary columns

The NH₂- and TpB-modified capillaries were first prepared for subsequent use (see SI for details). Pre-polymerization solutions of HTpBPa-F were injected into the TpB-modified capillaries. After sealing both ends, the capillaries were incubated at 90 °C for 72 h, then rinsed with EtOH, and dried with N₂ at 120 °C to obtain HTpBPa-F bonded capillary columns. The STpBPa-F and STpBPa bonded capillary columns were fabricated following the procedure for the HTpBPa-F column, but with some specific alterations. For the STpBPa-F bonded columns, the only change was a reduction in reaction time to 12 h. In contrast, the STpBPa bonded columns were synthesized using STpBPa pre-polymerization solution and reacted at room temperature.

### GC separation

Gas chromatographic separation of isomers was performed on a 2030 Plus GC (Shimadzu, Japan) with a flame ionization detector. High-purity N₂ (99.999%) was employed as the carrier gas. Prior to use, the

prepared capillary columns must be aged with the following temperature program: Ramp from 40 °C to 180 °C at a rate of 5 °C min$^{-1}$ and hold at 180 °C for 1 h with a $N_2$ flow rate of 1 mL min$^{-1}$. Chromatographic separation was conducted under optimal conditions.

## Data availability

All data are available within the article and supplementary files, or available from the corresponding authors on request. Source data are provided with this paper.

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

## Acknowledgements

The authors thank the Advance Analysis and Gene Sequencing Center of Zhengzhou University for Cryo-TEM analysis, and Dr. Lixia Yan from Analysis and Testing Center of Jiangnan University for TEM test. We also acknowledge the support from the National Natural Science Foundation of China (22422406, 22376082 and 22504045) and the "Fundamental Research Funds for the Central Universities".

## Author contributions

Zi-Rui Rao: Methodology, Formal analysis, Investigation, Writing—original draft. Xu-Qin Ran: Formal analysis and Methodology. Zhi-Quan Li and Yanlong Chen: Methodology. Hai-Long Qian: Conceptualization, Methodology, Writing—review & editing, Resources, Supervision, Funding acquisition. Xiu-Ping Yan: Writing—review & editing.

## Competing interests

The authors declare no competing interests.
