## [Transparent Peer Review file · Nature Communications]

Dual engineering of thermodynamics and kinetics in covalent organic frameworks for separation

Corresponding Author: Professor Hai-Long Qian

Version 0:

Reviewer comments:

Reviewer #1

(Remarks to the Author)

This manuscript reports an interesting example of engineering COF thermodynamics and kinetics via rational design of a new hollow fluorinated COF for chromatographic separation of halogenated isomers. Through introduction of trifluoromethyl groups, the proposed COF not only showed thermodynamic selectivity, but also exhibited a hollow structure for improving the separation kinetics for halogenated isomers. The paper describes detail design of the COF and separation results. Moreover, a thorough evaluation of a control COF and supporting theoretical calculations is given to rationalize the mechanism. Overall this could become a good contribution, certainly of interest to a wide readership.

Specific comments:

- (1) High-Resolution Mass Spectrometry (HRMS) is crucial for unambiguous characterization as it provides accurate mass measurements with precision typically to the fourth decimal place. HRMS should be applied in characterization of Pa-F.
- (2) Solid-state nuclear magnetic resonance spectroscopy is recommended to obtain more detailed composition of the fluorinated COF and F-free COF.
- (3) The authors should discuss or provide experimental data to clarify whether the separation occurs primarily on the surface or within the pores of the fluorinated COF. This distinction has significant implications for understanding the role of the COF porous structure in the separation.
- (4) To better illustrate the superiority of thermodynamic and kinetic regulation, the authors are encouraged to attempt the separation of more halogenated isomers.
- (5) In the discussion of thermodynamic mechanisms, the thermodynamic parameters (ΔH and ΔS) should be included in the discussion to gain a deeper understanding of the interactions between isomers and the proposed COFs.
- (6) A long-term stability is essential for the practical utility of column. Could the COF-based column be recovered after being used for a long while? A stabilization test should be conducted.

Minor revision:

- (1) Some units are presented inconsistently. For example, mL min⁻¹ vs cm²/s.
- (2) Abbreviations were not defined upon first use.
- (3) "HTpBP_a" in Figure 1 should be "HTpBP_a-F".
- (4) The reflection planes listed in the text do not match those labeled in Fig. 2.

Reviewer #2

(Remarks to the Author)

This manuscript reports an innovative perspective to tailor COFs from thermodynamics and kinetics for high-performance separation. They achieved this goal by the rational development of a hollow fluorinated COF. Fluorination can render COF thermodynamic selectivity for halogenated isomers, while a hollow structure significantly improved the separation kinetics. This work demonstrates the guidance for developing media for high-performance separation. The manuscript is properly organized, and the data are scientifically presented. I support the publication of this work after the following revisions.

- (1) Correct the label in Figure 1 from "HTpBP_a" to "HTpBP_a-F".
- (2) For effective comparison and to clearly highlight the growth of the COF-based capillary columns, SEM of the bare capillary is suggested to be provided.
- (3) I suggest the authors to investigate whether the approach can be extended to separate other common organic compounds, such as homologs, in addition to halogenated isomers.

- (4) To ensure a fair comparison, isomers must be separated under their optimal conditions for each respective column. Please make sure all the separation parameters for isomers were provided.
- (5) Experiments should be repeated at least three times to ensure reproducibility. The data in tables should include standard deviations, and the figures should incorporate error bars.
- (6) It is necessary to investigate not only the operational consistency of the columns but also their precision across different batches. The reproducibility and repeatability of the HTpBP_a-F capillary column need to be carefully investigated.
- (7) The current English writing still requires improvement. For example, "dual modulation" appears redundantly in the abstract section. The solid fluorinated COF named "STqPa-F" in the methods section is inconsistently referenced elsewhere.
- (8) Cite additional representative works on COF-based chromatographic stationary phases, such as J. Am. Chem. Soc. 2023, 145, 18956.

Reviewer #3

(Remarks to the Author)

This study proposes the "dual thermodynamic-dynamical engineering" strategy for the design of COF chromatographic stationary phases, which integrates thermodynamic selectivity tuning (via trifluoromethyl groups) with kinetic optimization (through a hollow structure) to achieve superior separation performance. The work presents a clear and innovative concept, representing a significant departure from the traditional single-factor optimization (of either thermodynamics or kinetics). The methodological design is relatively systematic, and the data fully support the conclusion.

Some specific review comments:

1. It is recommended to modify the title to include "for separation" because the core of this work is the dual engineering of the thermodynamics and kinetics governing the separation process. Suggestions: Dual engineering of thermodynamics and kinetics in covalent organic frameworks for separation.
2. The crystal The reflection planes assigned in PXRD characterization (Lines 126-127) did not match those labeled in Figure 2a. Please verify and correct this inconsistency.
3. The R_wp and R_p values from the rietveld refinement should be provided in the manuscript to support the validity of the structural model.
4. To ensure the reliability of the chromatographic data, the column's repeatability and stability must be established. Please provide the relevant experimental data.
5. To fully support the proposed separation mechanism, the relevant thermodynamic parameters (such as enthalpy or entropy changes) for the halogenated isomers should be investigated.
6. English expression needs revision:
 - Figure 1: "HTpBP_a"
 - Page 2, line 26: "indicating the successful dual modulation of dual modulation of thermodynamics and kinetics in the COF for separation",
 - Page 4, line 41: "researchers can tune their thermodynamics o to achieve separations of homologues".
 - Page 8, line 125 : "resulting no differences in their composition and structure".
 - The of Figure 4 caption: "(e) DBP (90 °C, 1.1 mL min⁻¹ of 209 N₂). (e) DCP (40 °C, 1.2 mL min⁻¹ of N₂)"
7. Some abbreviations are not defined in full upon first use.

Version 1:

Reviewer comments:

Reviewer #1

(Remarks to the Author)

This revised manuscript can be published now.

Reviewer #2

(Remarks to the Author)

The authors well addressed the reviewers' comments/questions in the revised version of this manuscript, and it can be accepted as is.

Reviewer #3

(Remarks to the Author)

The authors have completed revisions in accordance with the reviewers' comments. The manuscript is now ready for acceptance.

POINT-BY-POINT RESPONSE TO REVIEWERS

The comments and suggestions made by the reviewers are very helpful for us to revise our manuscript. We highly appreciate the reviewers for such constructive comments. Detail reply to the comments and suggestions is made below.

Response to Reviewer 1

Comment:

This manuscript reports an interesting example of engineering COF thermodynamics and kinetics via rational design of a new hollow fluorinated COF for chromatographic separation of halogenated isomers. Through introduction of trifluoromethyl groups, the proposed COF not only showed thermodynamic selectivity, but also exhibited a hollow structure for improving the separation kinetics for halogenated isomers. The paper describes detail design of the COF and separation results. Moreover, a thorough evaluation of a control COF and supporting theoretical calculations is given to rationalize the mechanism. Overall this could become a good contribution, certainly of interest to a wide readership.

Reply:

Thank you for your positive comments!

Comment 1

High-Resolution Mass Spectrometry (HRMS) is crucial for unambiguous characterization as it provides accurate mass measurements with precision typically to the fourth decimal place. HRMS should be applied in characterization of Pa-F.

Reply:

Good suggestion! We have characterized Pa-F using high-resolution mass spectrometry in the revised manuscript. The experimentally observed m/z value $[M+H]^+$ for Pa-F is 177.0645, which matches the theoretical value of 177.0561 (Please see the revised supplementary information, page S8, and Supplementary Fig. S3).

Comment 2

Solid-state nuclear magnetic resonance spectroscopy is recommended to obtain more detailed composition of the fluorinated COF and F-free COF.

Reply:

Thank you for your comment. The ^{13}C solid-state nuclear magnetic resonance (^{13}C -SNMR) spectroscopy of fluorinated COF and F-free COF was provided in the revised manuscript. Compared with the STpBPpa, HTpBPpa-F exhibited a carbon signal at 118 ppm in the ^{13}C -SNMR spectrum. This signal can be assigned to a carbon atom of C-F3 bond, confirming the successful introduction of the F groups (Please see the revised manuscript, page 8, first paragraph, lines 6-7; revised supplementary information, page S9, Supplementary Fig. S7).

Comment 3

The authors should discuss or provide experimental data to clarify whether the separation occurs primarily on the surface or within the pores of the fluorinated COF. This distinction has significant implications for understanding the role of the COF porous structure in the separation.

Reply:

Thank you for your comment. The molecular sizes of separated isomers range from 0.3 to 1.3 nm, all of which are smaller than the pore size of HTpBPpa-F. As a result, these molecules can diffuse into the pores, and the spatial effect of the pores can account for the separation mechanism. We have put the explanation in the revised manuscript (Please see revised manuscript, page 13, last line, page 14, lines 1-2 of the first paragraph; revised supplementary information, page S26, Supplementary Table S5).

Comment 4

To better illustrate the superiority of thermodynamic and kinetic regulation, the authors

are encouraged to attempt the separation of more halogenated isomers.

Reply:

Thank you for your comment. In light of your suggestion, three additional sets of halogenated isomers including tetrachlorobenzene (TeCB), bromochlorobenzene (BCB), and chloronitrobenzene (CNB) were separated in the revised manuscript. The HTpBPpa-F bonded capillary column showed higher performance in resolution and column efficiency for all the studied isomers, compared to the STpBPpa-F and STpBPpa capillary columns. Furthermore, the co-elution of CA, MoCB and TeCB isomers, as well as poor resolution of FA, 1,2-DCE and BCB isomers on benchmark commercial column HP-5 further indicate the great potential of HTpBPpa-F bonded capillary column as specific chromatography column for halogenated isomers (Please see revised manuscript, page 12, paragraph 2, lines 1-11; revised supplementary information, page S15, Supplementary Fig. S23, page S16, Supplementary Fig. S24, page S17, Supplementary Fig. S25).

Comment 5

In the discussion of thermodynamic mechanisms, the thermodynamic parameters (ΔH and ΔS) should be included in the discussion to gain a deeper understanding of the interactions between isomers and the proposed COFs.

Reply:

Thank you for your comment. In view of your suggestion, we have investigated the thermodynamic parameters for the separation of halogenated isomers in the revised manuscript. Negative ΔS values (-5.0 – -67.2 J mol⁻¹ K⁻¹) imply that all the separated halogenated isomers are 'locked' in a more ordered state while interacting with the HTpBPpa-F stationary phase during separation. As is known, enthalpy change (ΔH) stems from the collective effect of multiple noncovalent interactions. The negative ΔH typically indicates the presence of strong and specific interactions during the separation process (Langmuir 2026, 42, 1800–1819). So the negative ΔH values

observed in the chromatographic separation (-12.1 to -42.9 KJ mol⁻¹) demonstrate that some strong interactions indeed occur between HTpBP_a-F and the halogenated isomers. These discussions have been supplemented in the revised manuscript (Please see revised manuscript, page 14, the last paragraph, page 15, the first paragraph, Supplementary Table S9).

Comment 6

A long-term stability is essential for the practical utility of column. Could the COF-based column be recovered after being used for a long while? A stabilization test should be conducted.

Reply:

Thank you for your comment. Separation performance of CA isomers on the same HTpBP_a-F column at different time was compared to evaluate the separation stability. In contrast to the initial prepared HTpBP_a-F column, the column used for 120 days gave reductions of only 0.4%–2.6% and 0.1%–0.2% in column efficiency (N) and capacity factor (k), respectively. Moreover, even after the column was subjected to 10 continuous programmed temperature cycles from 50 to 300°C at 2°C min⁻¹, the N and k for the CA isomers showed no significant change, demonstrating the excellent long-term stability of the prepared HTpBP_a-F column. (Please see revised manuscript, page 18, the last paragraph; revised supplementary information, page S20, Supplementary Fig. S30, page S32, Supplementary Table S12).

Comment 7

Some units are presented inconsistently. For example, mL min⁻¹ vs cm²/s.

Reply:

Thank you for your comment. “The cm²/s” is now revised to “cm² s⁻¹”. We also checked all the manuscript to ensure all the units are presented consistently (Please see revised manuscript, page 17, the first paragraph, Lines 7-8).

Comment 8

Abbreviations were not defined upon first use.

Reply:

Thank you for your comment. We have checked the entire manuscript to ensure that all abbreviations are defined upon first use (Please see revised manuscript, page 2, line 3 of abstract, page 8, the first line).

Comment 9

“HTpBPa” in Figure 1 should be “HTpBPa-F”.

Reply:

Thank you for your comment. The “HTpBPa” in Figure 1 have been revised to “HTpBPa-F” (Please see revised manuscript, page 7, Fig 1).

Comment 10

The reflection planes listed in the text do not match those labeled in Fig. 2.

Reply:

Thank you for your correction. PXRD pattern showed peaks at 2.9°, 5.0°, 5.7°, and 7.6° (Fig. 2a), corresponding to reflections from the (100), (110), (020), and (120) planes, respectively. The reflection planes labeled in Fig. 2a was corrected in the revised manuscript (Please see revised manuscript, page 10, Fig. 2a).

Response to Reviewer 2

Comment:

This manuscript reports an innovative perspective to tailor COFs from thermodynamics and kinetics for high-performance separation. They achieved this goal by the rational development of a hollow fluorinated COF. Fluorination can render COF thermodynamic selectivity for halogenated isomers, while a hollow structure significantly improved the separation kinetics. This work demonstrates the guidance for developing media for high-performance separation. The manuscript is properly organized, and the data are scientifically presented. I support the publication of this work after the following revisions.

Reply:

Thank you for your positive comments!

Comment 1

Correct the label in Figure 1 from "HTpBPa" to "HTpBPa-F".

Reply:

Thank you for your correction. The "HTpBPa" in Figure 1 have been revised to "HTpBPa-F" (Please see revised manuscript, page 7, Fig 1).

Comment 2

For effective comparison and to clearly highlight the growth of the COF-based capillary columns, SEM of the bare capillary is suggested to be provided.

Reply:

Thank you for your constructive suggestion. We have provided SEM images of the bare capillary in the revised manuscript. SEM images confirmed the smooth surface of bare capillary. In contrast, the inner walls of the COFs capillary columns were uniformly coated with dense particles of HTpBPa-F, STpBPa-F, and STpBPa (Please see revised manuscript, page 10, the first paragraph, lines 7-9; revised supplementary information,

page S14, Supplementary Fig. S20).

Comment 3

I suggest the authors to investigate whether the approach can be extended to separate other common organic compounds, such as homologs, in addition to halogenated isomers.

Reply:

Thank you for your comment. In view your suggestion, we have separated the homologs including n-alkanes and n-alcohols. The HTpBPa-F bonded capillary column exhibited baseline separation of n-alkanes (n-pentane to n-dodecane) and n-alcohols (methanol to 1-octanol) under constant temperature. As the number of carbon atoms increases, the boiling points of these homologues significantly increase, resulting in longer separation times. If the separation performed with temperature-programmed, the separation time was evidently decreased. More analytes (n-pentane to n-tetradecane) can be baseline-separated. This highlights the broad practical utility of HTpBPa-F bonded capillary column. These contents have been supplemented in the revised manuscript (Please see revised manuscript, page 12, the last four lines, supplementary information, page S18, Supplementary Fig. S26).

Comment 4

To ensure a fair comparison, isomers must be separated under their optimal conditions for each respective column. Please make sure all the separation parameters for isomers were provided.

Reply:

Thank you for your comment. We have supplemented the statement of “separation conditions were optimized to achieve the best separation of the isomers.” in the revised manuscript (Please see revised supplementary information, page 15, Supplementary Fig. S23 caption, page S16, Supplementary Fig. S24 caption, page S17,

Supplementary Fig. S25 caption).

Comment 5

Experiments should be repeated at least three times to ensure reproducibility. The data in tables should include standard deviations, and the figures should incorporate error bars.

Reply:

Thank you for your constructive comment. We have repeated all key experiments at least three times. Standard deviations in all relevant tables and error bars in the figures were supplemented in the revised manuscript (Please see revised supplementary information, page S19, Supplementary Fig. S28, page S24, Supplementary Table S3, page S25, Supplementary Table S4, page S27, Supplementary Table S6, page S28, Supplementary Table S7, page S29, Supplementary Table S8, page S30, Supplementary Table S9, page S31, Supplementary Table S10, page S32, Supplementary Table S12).

Comment 6

It is necessary to investigate not only the operational consistency of the columns but also their precision across different batches. The reproducibility and repeatability of the HTpBPpa-F capillary column need to be carefully investigated.

Reply:

Thank you for your comment. Both the operational consistency and inter-batch precision were further investigated to assess the repeatability of HTpBPpa-F capillary column. Specifically, the relative standard deviations (RSDs) for the retention time and peak area of CA isomers with a same HTpBPpa-F capillary column were 0.11%-0.12% and 2.8%-3.2% for eight consecutive runs, respectively. For five runs repeated across different days, the corresponding RSDs were 0.10–0.11% and 5.8–6.1%, respectively. These results clearly demonstrate the high repeatability of the HTpBPpa-F capillary

column. In addition, excellent reproducibility was also confirmed by the RSDs of 0.11%-0.22% (retention time) and 6.8%-7.4% (peak area) for CA isomers obtained using columns from three different batches (Please see revised manuscript, page 19, the first paragraph; revised supplementary information, page S32, Supplementary Table S13).

Comment 7

The current English writing still requires improvement. For example, "dual modulation" appears redundantly in the abstract section. The solid fluorinated COF named "STqPa-F" in the methods section is inconsistently referenced elsewhere.

Reply:

Thank you for your comment. We have carefully revised the language throughout the manuscript to ensure clear and professional communication of all findings.

Comment 8

Cite additional representative works on COF-based chromatographic stationary phases, such as J. Am. Chem. Soc. 2023, 145, 18956.

Reply:

Thank you for your comment. The suggested reference was cited in the revised manuscript (Please see revised manuscript, page 23, reference 13).

Response to Reviewer 3

This study proposes the "dual thermodynamic-dynamical engineering" strategy for the design of COF chromatographic stationary phases, which integrates thermodynamic selectivity tuning (via trifluoromethyl groups) with kinetic optimization (through a hollow structure) to achieve superior separation performance. The work presents a clear and innovative concept, representing a significant departure from the traditional single-factor optimization (of either thermodynamics or kinetics). The methodological design is relatively systematic, and the data fully support the conclusion.

Reply:

Thanks very much for your valuable and positive comments!

Comment 1

It is recommended to modify the title to include "for separation" because the core of this work is the dual engineering of the thermodynamics and kinetics governing the separation process. Suggestions: Dual engineering of thermodynamics and kinetics in covalent organic frameworks for separation.

Reply:

Good suggestion. In light of your suggestion, the title was revised to "Dual engineering of thermodynamics and kinetics in covalent organic frameworks for separation" in the revised manuscript.

Comment 2

The crystal reflection planes assigned in PXRD characterization (Lines 126-127) did not match those labeled in Figure 2a. Please verify and correct this inconsistency.

Reply:

Thank you for your correction. Thank you for your correction. PXRD pattern showed peaks at 2.9°, 5.0°, 5.7°, and 7.6° (Fig. 2a), corresponding to reflections from the (100), (110), (020), and (120) planes, respectively. The reflection planes labeled in Fig. 2a

was corrected in the revised manuscript (Please see revised manuscript, page 10, Fig. 2a).

Comment 3

The Rwp and Rp values from the rietveld refinement should be provided in the manuscript to support the validity of the structural model.

Reply:

Thank you for your comment. We have added the Rwp and Rp values obtained from the Rietveld refinement to the revised manuscript. The low Rwp and Rp values for HTpBPa-F (7.37% and 5.48%) and STpBPa (5.47% and 4.21%) indicate the reliability and validity of the refined structural models (Please see revised manuscript, page 9, Fig 2 caption; revised supplementary information, page S11, Supplementary Fig. S13 caption).

Comment 4

To ensure the reliability of the chromatographic data, the column's repeatability and stability must be established. Please provide the relevant experimental data.

Reply:

Thank you for your comment. Both the operational consistency and inter-batch precision were further investigated to assess the repeatability of the HTpBPa-F capillary column. Specifically, the relative standard deviations (RSDs) for the retention time and peak area of CA isomers with a same HTpBPa-F capillary column were 0.11%-0.12% and 2.8%-3.2% for eight consecutive runs, respectively. For five runs repeated across different days, the corresponding RSDs were 0.10–0.11% and 5.8–6.1%, respectively. These results clearly demonstrate the high repeatability of the HTpBPa-F capillary column. In addition, excellent reproducibility was also confirmed by the RSDs of 0.11%-0.22% (retention time) and 6.8%-7.4% (peak area) for CA isomers obtained using columns from three different batches (Please see revised

manuscript, page 19, the first paragraph; revised supplementary information, page S32, Supplementary Table S13).

Separation performance of CA isomers on the same HTpBPpa-F column at different time was compared to evaluate the separation stability. In contrast to the initial prepared HTpBPpa-F column, the column used for 120 days gave reductions of only 0.4%–2.6% and 0.1%–0.2% in column efficiency (N) and capacity factor (k), respectively. Moreover, even after the column was subjected to 10 continuous programmed temperature cycles from 50 to 300°C at 2°C min⁻¹, the N and k for the CA isomers showed no significant change, demonstrating the excellent long-term thermal stability of the prepared HTpBPpa-F column (Please see revised manuscript, page 18, the last paragraph; revised supplementary information, page S20, Supplementary Fig. S31, page S32, Supplementary Table S12).

Comment 5

To fully support the proposed separation mechanism, the relevant thermodynamic parameters (such as enthalpy or entropy changes) for the halogenated isomers should be investigated.

Reply:

Thank you for your comment. In view of your suggestion, we have investigated the thermodynamic parameters for the separation of halogenated isomers in the revised manuscript. Negative ΔS values (-5.0 – -67.2 J mol⁻¹ K⁻¹) imply that all the separated halogenated isomers are 'locked' in a more ordered state while interacting with the HTpBPpa-F stationary phase during separation. As is known, enthalpy change (ΔH) stems from the collective effect of multiple noncovalent interactions. The negative ΔH typically indicates the presence of strong and specific interactions during the separation process (Langmuir 2026, 42, 1800–1819). So the negative ΔH values observed in the chromatographic separation (-12.1 to -42.9 KJ mol⁻¹) demonstrate that some strong interactions indeed occur between HTpBPpa-F and the halogenated

isomers. These discussions have been supplemented in the revised manuscript (Please see revised manuscript, page 14, the last paragraph, page 15, the first paragraph, revised supplementary information, Supplementary Table S9).

Comment 6

English expression needs revision:

—Figure 1: “HTpBP_a”

—Page 2, line 26: “indicating the successful dual modulation of dual modulation of thermodynamics and kinetics in the COF for separation”,

—Page 4, line 41: “researchers can tune their thermodynamics o to achieve separations of homologues”.

—Page 8, line 125: “resulting no differences in their composition and structure”.

—The of Figure 4 caption: “(e) DBP (90 °C, 1.1 mL min⁻¹ of 209 N₂). (e) DCP (40 °C, 1.2 mL min⁻¹ of N₂)”.

Reply:

Thank you for your comment. “HTpBP_a” was revised into “HTpBP_a-F” (Please see revised manuscript, page 7, Fig 1). The repetitive content “dual modulation of” and redundant words “o” were deleted. The “resulting no differences” was revised to “resulting in no differences” (Please see revised manuscript, page 7, the first paragraph, line 4); “(e) DCP” was revised to “(f) DCP” (Please see revised manuscript, page 13, Fig 4 caption).

Comment 7

Some abbreviations are not defined in full upon first use.

Reply:

Thank you for your comment. We have checked the entire manuscript to ensure that all abbreviations are defined upon first use (Please see revised manuscript, page 2, line 3 of abstract, page 8, the first line).